# Bigmouth Buffalo *Ictiobus cyprinellus* sets freshwater teleost record as improved age analysis reveals centenarian longevity

Alec R. Lackmann[1], Allen H. Andrews[2], Malcolm G. Butler[3], Ewelina S. Bielak-Lackmann[3] & Mark E. Clark[3]

Understanding the age structure and population dynamics of harvested species is crucial for sustainability, especially in fisheries. The Bigmouth Buffalo (*Ictiobus cyprinellus*) is a fish endemic to the Mississippi and Hudson Bay drainages. A valued food-fish for centuries, they are now a prized sportfish as night bowfishing has become a million-dollar industry in the past decade. All harvest is virtually unregulated and unstudied, and Bigmouth Buffalo are declining while little is known about their biology. Using thin-sectioned otoliths and bomb-radiocarbon dating, we find Bigmouth Buffalo can reach 112 years of age, more than quad-rupling previous longevity estimates, making this the oldest known freshwater teleost (~12,000 species). We document numerous populations that are comprised largely (85–90%) of individuals over 80 years old, suggesting long-term recruitment failure since dam construction in the 1930s. Our findings indicate Bigmouth Buffalo require urgent attention, while other understudied fishes may be threatened by similar ecological neglect.

[1] Department of Biological Sciences, North Dakota State University, Environmental and Conservation Sciences Program, 1340 Bolley Drive, Fargo, ND 58102, USA. [2] Department of Oceanography, University of Hawaii at Manoa, 1000 Pope Road, Honolulu, HI 96822, USA. [3] Department of Biological Sciences, North Dakota State University, Fargo, ND 58102, USA. Correspondence and requests for materials should be addressed to A.R.L. (email: alec.lackmann@ndsu.edu)

The Bigmouth Buffalo (*Ictiobus cyprinellus*) is one of the largest freshwater fishes endemic to North America, reaching lengths exceeding 1.25 m and body masses >36 kg[1]. Indeed, it is the largest of all catostomids (Cypriniformes: Catostomidae). Bigmouth Buffalo are also unique as the only catostomid with a terminal mouth and planktivorous, filter-feeding tendencies. All other catostomids are benthivorous[2]. The life history of *I. cyprinellus* was described previously as fast-paced[2], despite apparent conflicting evidence from two studies reporting failure of some mature females to spawn every year[2,3]. One study reported a maximum estimated age of 26 years[4], but previous reports suggested a younger maximum age (10–20 years), and reproductive maturity occurring as early as the first year of growth[2,5,6]. This exclusively freshwater species inhabits shallow (<4 m) warm-water lakes and pond-like areas of rivers, and is tolerant of eutrophication and high turbidity[1,2]. Shallow habitats are not typically associated with a long lifespan[7,8].

Bigmouth Buffalo have been important to human cultures in North America. Several lake names in Minnesota use the word *niigijiikaag*, the Ojibwe (a regional Native American tribe) name for buffalofish (Klimah, C., Minnesota Department of Natural Resources Fisheries Biologist, 2018, personal communication). Other Minnesota lakes and one county were named *Kandiyohi* by the Dakota tribe, meaning "where the buffalofish come." In addition, the city of Buffalo, MN is named after this species[9]. In 1804, Lewis and Clark harvested buffalofish in Nebraska[10] and they have been of commercial importance since the 1800s[11,12]. This fishery is valued in the 21st Century at over 1 million USD per year in the Upper Mississippi Basin alone[13]. Despite its value, Bigmouth Buffalo have become increasingly misunderstood over the past century as they became commonly categorized as a "rough fish." This imprecise term is used in much of the USA to lump many endemic, traditionally nongame fishes, along with unwanted invasive fishes, for purposes of harvest regulation[14]. This pejorative designation has led to the misconception by the general public of Bigmouth Buffalo as an "invasive species" or "a carp," encouraging its persecution as a sacrificial or unimportant species. Contrary to this treatment in the USA, Bigmouth Buffalo were given Special Concern status in the Hudson Bay drainage of Canada in 1989 by the Committee on the Status of Endangered Wildlife in Canada following documented population decline concomitant with increases in invasive Common Carp (*Cyprinus carpio*)[15]. Bigmouth Buffalo serve as a competitor to the invasive Bighead Carp (*Hypophthalmichthys nobilis*) and Silver Carp (*H. molitrix*)[5,16–23], as well as the invasive Common Carp[2,24], thus these three invasive species pose threats in addition to overharvest. Hence there is a basis for considering Bigmouth Buffalo as an ecological asset, and reason for concern about declining populations of Bigmouth Buffalo that have been documented in the northern parts of their range, including Canada, Minnesota, and North Dakota[1,15]. Unfortunately, other North American catostomids may also warrant such concern, with 42 out of 76 species already classified as endangered, threatened, vulnerable, or extinct, according to a recent synthesis on the conservation status of this family[25].

Current harvest of Bigmouth Buffalo is largely unregulated. This is partly because Bigmouth Buffalo have long been unpopular with recreational anglers, as these pelagic filter-feeders rarely take a baited hook or lure and thus are seldom caught by hook-and-line. However, legislative changes in the past decade coincide with a sharp increase in the popularity of bowfishing[26]. Across the USA, bowfishing is now permitted at night; archers can shoot "rough fish" with a bow and arrow under powerful lights, despite little to no regulation or study of this new harvest method[27]. While Bigmouth Buffalo and several other endemic taxa have become prized catches for bowfishers[28], angler harvest of Bigmouth Buffalo in the USA is currently unregulated in 19 of the 22 states to which they are endemic, where recreational anglers can harvest unlimited numbers. Exceptions include Missouri and Louisiana with established take limits, and in Pennsylvania where Bigmouth Buffalo are considered endangered and are illegal to possess[14]. Furthermore, commercial anglers face no limits on the total number of Bigmouth Buffalo harvested in any U.S. state except Pennsylvania, and fish size restrictions on commercial harvest exist only in Louisiana and Mississippi[14]. Given the largely unregulated harvest of this ecologically important and historically valued endemic fish, it is crucial to validate its life history characteristics.

We use otoliths (earstones) to estimate demographic characteristics of *I. cyprinellus* collected from 12 populations in two major drainages in Minnesota, and annulus counts on thin-sectioned otoliths to estimate fish age. The validity of these age estimates is tested using bomb radiocarbon ($^{14}$C) dating, a method that relies on bomb-produced $^{14}$C from atmospheric thermonuclear testing in the 1950s and 1960s as a time-specific marker[29]. These validated age-at-length data are used to describe Bigmouth Buffalo growth characteristics and age-at-maturity, which differ by an order of magnitude from previously published work on this species. We also report on novel age-related external markings that aid individual recognition and mark-recapture, as well as provide a non-lethal means of age estimation.

## Results

**Age analysis.** During the years 2016 to 2018, we estimated the age for 386 Bigmouth Buffalo by counting annuli in thin-sectioned otoliths (Fig. 1), currently the most reliable method for age estimation of teleost fishes[30]. We investigated the three pairs of otoliths (sagittae, lapilli, and asterisci) for growth zone structure that could be interpreted as annual. Specimens used in this study came from 12 populations spanning two drainages in Minnesota: the Red River Basin of the North ($n = 257$), a part of the Hudson Bay watershed; and the Mississippi River Basin ($n = 129$). From the Red River Basin, 224 fish came from eight lakes along a 26-km reach of the Pelican River Basin in Otter Tail County: Prairie ($n = 7$), North Lida ($n = 42$), Crystal ($n = 59$), Rush ($n = 37$), Lizzie ($n = 46$), Fish ($n = 1$), Big Pelican ($n = 19$), and Little Pelican ($n = 13$). The remaining 33 Red River Basin specimens came from the Otter Tail River below Orwell Dam. From the Mississippi River Basin, we obtained 129 Bigmouth Buffalo from three lakes: Minnetaga (Kandiyohi County; $n = 66$) along the Crow River; plus Artichoke (Big Stone and Swift County; $n = 52$), and Ten Mile (Otter Tail County; $n = 11$) along the Minnesota River.

Age estimates from thin-sectioned otoliths (lapilli and asterisci) were validated using bomb $^{14}$C dating (see Methods). Both of these otoliths were validated for age analysis by the strong agreement between the quantified annuli across all ages (Fig. 2). Furthermore, many of the otoliths provided for bomb $^{14}$C dating were scored exclusively using an asteriscus ($n = 7$; Table 1), even though a lapillus (read only to age two for core extraction) was used for bomb radiocarbon analyses, and expected $^{14}$C results were obtained in all cases. All samples analyzed revealed $^{14}$C values that were consistent with the birth years generated from annulus counts (Table 1, Fig. 3), when compared to expected $^{14}$C levels associated with the bomb $^{14}$C reference records that are available for freshwater bodies of North America (Fig. 4). One exception was the sole Mississippi River Basin sample; this anomaly was likely a basin effect as all other samples were from the Red River Basin. Specifically, samples extracted from annuli representing birth years in pre-bomb, rise, and post-peak decline periods resulted in $^{14}$C values that were consistent with freshwater $^{14}$C reference records (Table 1, Fig. 3). All fish estimated to have hatched prior to atmospheric nuclear testing had otolith

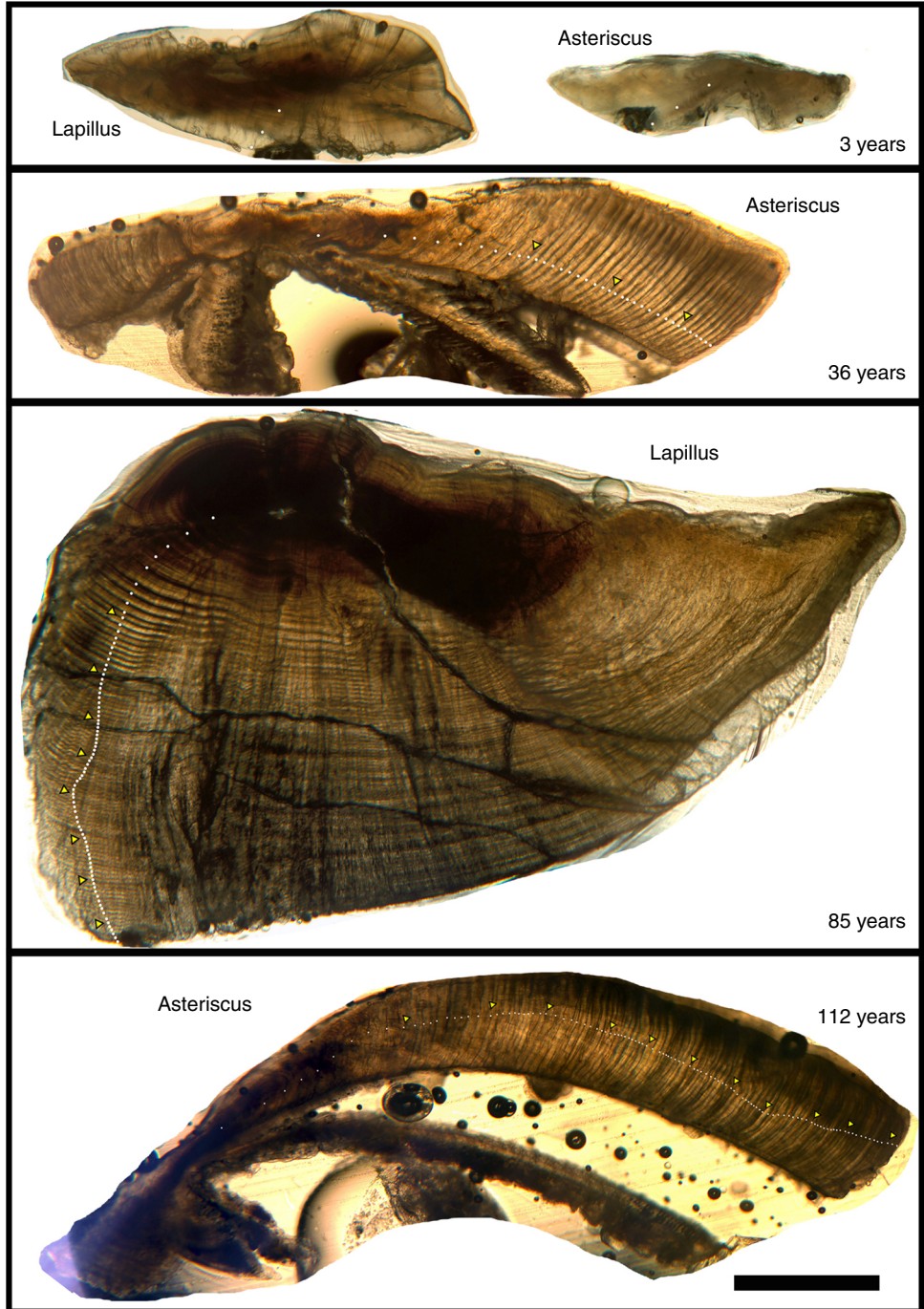

**Fig. 1** Thin-sectioned otoliths. Thin-sectioned lapillus and asteriscus otoliths from four Bigmouth Buffalo (*Ictiobus cyprinellus*) with age estimates ranging from 3, 36, 85, and 112 years at the time of collection. White dots indicate annual growth bands and yellow triangles decade counts. All otolith composite images set to same scale bar = 1 mm. Note the well-defined annuli

core radiocarbon values indicative of a minimum age that exceeded 60 years, as expected. These pre-bomb otolith core $^{14}$C values were very consistent through time (1926–1938) with a mean of −39.4‰ ± 3.3 SD (Fig. 3). The one available rise period specimen (ICCY-09) had a diagnostic $\Delta^{14}$C value of 134.9‰ at a birth year of 1960, for a validated age of 58 ± 1–2 years. Variation from this rise time should be within 2 years because of the rapid and time-specific increase in $^{14}$C, assuming the regional hydrogeology is similar to the available $^{14}$C references. Younger fish were also consistent with the general expectation for the bomb $^{14}$C peak and subsequent decline period, to the

extent that specimens were available. No fish with birth years near the expected peak of bomb-produced $^{14}$C (~1965) were available, but specimens with birth years in the 1970s through to the 2000–2010s showed a temporal $^{14}$C decline consistent with the declining trend shown by reference $^{14}$C records.

Radial samples that covered multiple years of growth for each of two Bigmouth Buffalo were also consistent with expected time-specific $^{14}$C levels, and supported annulus-count ages of 90 and 92 years (Table 1, Fig. 3). These fish lived their first three decades prior to atmospheric bomb testing and carbonate samples from otolith locations corresponding to those pre-bomb years were

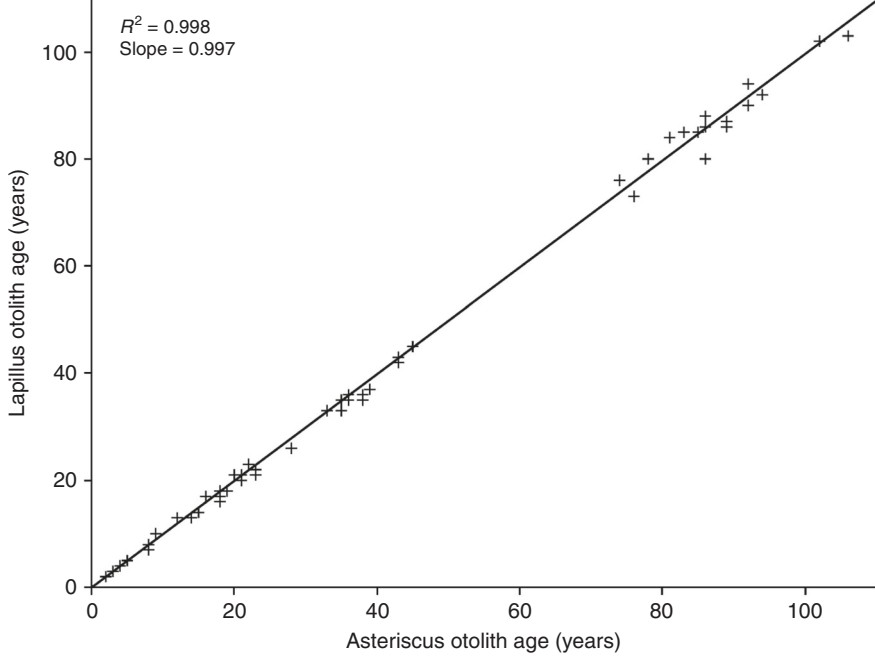

**Fig. 2** Annulus counts of lapillus vs. asteriscus otoliths from the same fish. Comparison of age readings made by the primary reader from thin-sectioned lapillus vs. asteriscus otoliths of the same Bigmouth Buffalo (*Ictiobus cyprinellus*) specimens ($n = 72$); Pearson Correlation Coefficient = 0.999, $p < 0.0001$; Paired *t*-test: t-Ratio = −1.813, $p > 0.05$, mean difference = −0.306. The linear regression slope estimate (0.997; 95% CI: [0.986, 1.007]) includes 1.000, and thus was statistically isometric. Hence, either the lapillus or asteriscus otolith can be used to consistently age Bigmouth Buffalo with precision and accuracy

## Table 1 Specimen data for 28 *Ictiobus cyprinellus* samples analyzed using bomb $^{14}$C dating

| Sample ID Sample mass (mg) | TL (cm) | Sex | Collection year | Otolith aged | Age in years | Formation year | Formation period | Δ$^{14}$C(‰) | | Error (‰) |
|---|---|---|---|---|---|---|---|---|---|---|
| ICCY-01 | 87.3 | ♀ | 2017 | A, L | 43 | 1974 | Post | 36.8 | 3.1 | 0.870 |
| ICCY-02 | 87.9 | ♀ | 2017 | A | 86 | 1931 | Pre-bomb | −42.3 | 2.2 | 1.384 |
| ICCY-03 | 96.9 | ♀ | 2017 | A | 85 | 1932 | Pre-bomb | −38.6 | 3.3 | 0.949 |
| ICCY-04 | 76.6 | ♂ | 2017 | A | 87 | 1930 | Pre-bomb | −38.8 | 3.3 | 0.873 |
| ICCY-06 | 48.1 | ♂ | 2018 | L | 13 | 2005 | Post | 47.6 | 4.7 | 1.098 |
| ICCY-07 | 87.3 | ♀ | 2018 | L | 80 | 1938 | Pre-bomb | −34.6 | 3.3 | 0.894 |
| ICCY-08 | 43.1 | ♂ | 2018 | L | 3 | 2015 | Post | −20.3 | 4.6 | 0.978 |
| ICCY-09 | 69.1 | ♂ | 2018 | L | 58 | 1960 | Rise | 134.9 | 4.9 | 0.873 |
| ICCY-10 | 84.0 | ♀ | 2018 | L | 42 | 1976 | Post | 214.3 | 3.3 | 0.886 |
| ICCY-13 | 88.3 | ♀ | 2018 | A | 43 | 1975 | Post | 201.3 | 4.8 | 0.887 |
| ICCY-15C | 103.1 | ♀ | 2018 | A, L | 92 | 1926 | Pre-bomb | −42.9 | 2.6 | 0.977 |
| ICCY-15R1 | | | | | 90 | 1928.2 | Pre-bomb | −64.1 | 3.2 | 0.753 |
| ICCY-15R2 | | | | | 84 | 1934.6 | Pre-bomb | −84.7 | 3.5 | 0.772 |
| ICCY-15R3 | | | | | 77 | 1941.0 | Pre-bomb | −77.8 | 3.2 | 0.804 |
| ICCY-15R4 | | | | | 67 | 1951.2 | 78% PB, 22% R[a] | −51.0 | 3.7 | 0.851 |
| ICCY-17 | 82.3 | ♂ | 2018 | A | 84 | 1934 | Pre-bomb | −42.4 | 3.0 | 0.956 |
| ICCY-21 | 68.5 | ♂ | 2018 | A | 90 | 1928 | Pre-bomb | −35.0 | 2.5 | 1.026 |
| ICCY-25 | 63.9 | ♂ | 2018 | A | 18 | 2000 | Post | 47.8 | 3.1 | 1.002 |
| ICCY-27C | 93.0 | ♀ | 2018 | A, L | 90 | 1928 | Pre-bomb | −40.7 | 2.7 | 0.875 |
| ICCY-27R1 | | | | | 87 | 1930.8 | Pre-bomb | −51.6 | 3.7 | 0.685 |
| ICCY-27R2 | | | | | 81 | 1936.5 | Pre-bomb | −59.2 | 3.6 | 0.692 |
| ICCY-27R3 | | | | | 75 | 1942.2 | Pre-bomb | −53.2 | 4.4 | 0.641 |
| ICCY-27R4 | | | | | 69 | 1948.7 | 85% PB, 15% R[a] | −47.8 | 3.6 | 0.623 |
| ICCY-27R5 | | | | | 59 | 1959.0 | 39% PB, 32% R, 29% P[a] | 62.1 | 4.0 | 0.613 |
| ICCY-27R6 | | | | | 51 | 1966.7 | 52% R, 48% P[a] | 167.2 | 4.3 | 0.586 |
| ICCY-27R7 | | | | | 42 | 1976.2 | 6% R, 94% P[a] | 193.5 | 4.2 | 0.700 |
| ICCY-27R8 | | | | | 30 | 1988.2 | Post | 110.8 | 3.7 | 0.674 |
| ICCY-27R9 | | | | | 19 | 1998.7 | Post | 44.2 | 4.4 | 0.560 |

Age estimates represent the time between the mean year of carbonate deposition to the collection year ( = age of fish for core samples). Carbon isotope ratios were determined from the micromill extracted sample. All specimens were from the Hudson Bay drainage except ICCY-01 (Mississippi River drainage) – specimen $^{14}$C measurement not plotted (Fig. 3) due to apparent differences in the drainage bomb $^{14}$C signal
*TL* total length, *A* asteriscus, *L* lapillus
[a]Extracted samples spanning greater than one formation period are given estimated percentages; others represent 100% of the sample. Estimated contribution based on an area overlay analysis of extraction vs. the growth zone structure. Pre-bomb (PB) < 1955, Rise (R) = 1955–1965, and Post (P) = all years after 1965. (More information is reported in Table 2).

**Table 2 Additional sample data for the radiocarbon analyses of Bigmouth Buffalo**

| Sample ID (NOSAMS ID) | F$^{14}$C | Error (‰) | δ$^{13}$C (‰) |
|---|---|---|---|
| ICCY01 (OS-143670) | 1.0398 | 0.0031 | −11.8 |
| ICCY02 (OS-143689) | 0.9555 | 0.0022 | −7.6 |
| ICCY03 (OS-143681) | 0.9593 | 0.0033 | −9.1 |
| ICCY04 (OS-143682) | 0.9589 | 0.0033 | −7.8 |
| ICCY06 (OS-143688) | 1.0546 | 0.0047 | 1.9 |
| ICCY07 (OS-143675) | 0.9640 | 0.0033 | −8.7 |
| ICCY08 (OS-143666) | 0.9874 | 0.0046 | −11.3 |
| ICCY09 (OS-143671) | 1.1363 | 0.0049 | −10.9 |
| ICCY10 (OS-143668) | 1.2181 | 0.0033 | 1.9 |
| ICCY13 (OS-143669) | 1.2049 | 0.0048 | −9.9 |
| ICCY15C (OS-143677) | 0.9543 | 0.0026 | −7.6 |
| ICCY15R1 (OS-143678) | 0.9334 | 0.0032 | −7.5 |
| ICCY15R2 (OS-143679) | 0.9136 | 0.0035 | −6.3 |
| ICCY15R3 (OS-143680) | 0.9212 | 0.0032 | −6.9 |
| ICCY15R4 (OS-144779) | 0.9491 | 0.0037 | n.m. |
| ICCY17 (OS-143676) | 0.9557 | 0.0030 | −8.2 |
| ICCY21 (OS-143690) | 0.9624 | 0.0025 | −7.9 |
| ICCY25 (OS-143667) | 1.0542 | 0.0031 | −11.6 |
| ICCY27C (OS-143672) | 0.9569 | 0.0027 | −6.8 |
| ICCY27R1 (OS-143673) | 0.9462 | 0.0037 | −7.6 |
| ICCY27R2 (OS-143674) | 0.9393 | 0.0036 | −6.7 |
| ICCY27R3 (OS-144772) | 0.9459 | 0.0044 | −6.0 |
| ICCY27R4 (OS-144773) | 0.9521 | 0.0036 | −5.6 |
| ICCY27R5 (OS-144774) | 1.0633 | 0.0040 | −5.8 |
| ICCY27R6 (OS-144775) | 1.1696 | 0.0043 | −5.8 |
| ICCY27R7 (OS-144776) | 1.1973 | 0.0042 | −5.5 |
| ICCY27R8 (OS-144777) | 1.1159 | 0.0037 | −5.8 |
| ICCY27R9 (OS-144778) | 1.0504 | 0.0044 | −5.9 |

This includes the NOSAMS processing number, reported F$^{14}$C values with error, and robust δ$^{13}$C measurements. These values are consistent with other freshwater otolith records[35,45]. Low δ$^{13}$C values indicate isotopic fractionation is a factor[58] and may also indicate that uptake of $^{14}$C-depleted carbon sources drives the observed pre-bomb $^{14}$C variation in Bigmouth Buffalo otoliths
*n.m.* not measured

within expected $^{14}$C levels from previous freshwater studies, but also may have set a new baseline for pre-bomb levels. The most comprehensive extraction series ($n = 10$) from ICCY-27 provided the most robust $^{14}$C data. Measured $^{14}$C values aligned strongly with $^{14}$C reference data through the last six decades of the 90-year-old lifespan of this fish. The core and four sequential radial samples were all pre-bomb, with the $^{14}$C rise occurring as predicted (based on annulus counts) in the late 1950s. The subsequent peak and declining $^{14}$C values were also consistent with reference records. While the pre-bomb otolith core $^{14}$C values were very consistent (−39.4‰ ± 3.3 SD) among individuals, pre-bomb radial samples tended to be more depleted with a mean of −61.2‰ ± 13.5 SD (Fig. 3), which is likely associated with a minor ontogenetic shift in habitat to slightly deeper, more $^{14}$C-depleted, waters after the first 1–2 years of growth. Overall, the findings from bomb $^{14}$C dating using both core and radial samples indicate the age reading protocol is valid and that age estimates approaching and exceeding 110 years are well-supported.

In all but one case (223 of 224), Bigmouth Buffalo from the Pelican River Basin were older than the previously reported maximum age of 26 years[4], with 186 of 224 individuals exceeding 75 years of age (1906–1942 year-classes). The remaining 39 Pelican River Basin fish ranged 18–49 years old (1969–2000 year-classes; Fig. 5). The five oldest Bigmouth Buffalo all exceeded 100 years, with the oldest estimated at 112 years old (Fig. 1). The 33 Red River Basin fish from Orwell Dam on the Otter Tail River ranged 3–80 years (1938–2015 year-classes; Fig. 5). In the Mississippi River Basin, two of the sampled lakes were subject to commercial harvest. At Artichoke Lake on the Minnesota River, 52 Bigmouth Buffalo selected from the harvest ranged 2–43 years (Fig. 5). At Lake Minnetaga (Crow River), we obtained 66

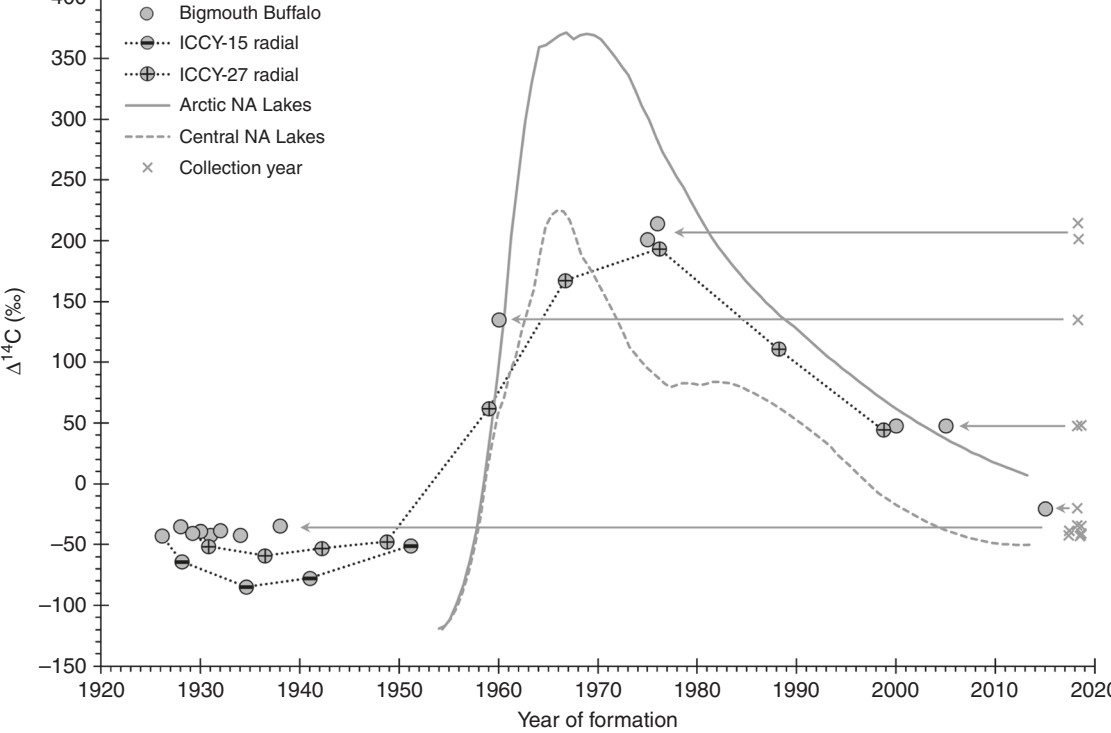

**Fig. 3** Age validation of Bigmouth Buffalo (*Ictiobus cyprinellus*). Radiocarbon (Δ$^{14}$C) measurements for the estimated year of otolith core formation of 14 Bigmouth Buffalo collected from the Hudson Bay drainage in 2017–2018. These specimens were estimated to be 3 to 92 years old (birth years of 1926–2015) from annulus counts. Reference curves for bomb-produced $^{14}$C were generated from the only thorough records from otoliths of freshwater fishes. Note the rise of bomb-produced $^{14}$C is similar across these regions of North America. The two sets of connected samples represent radial extraction series for two specimens sampled from the otolith core, through a sequential set of estimated formation dates

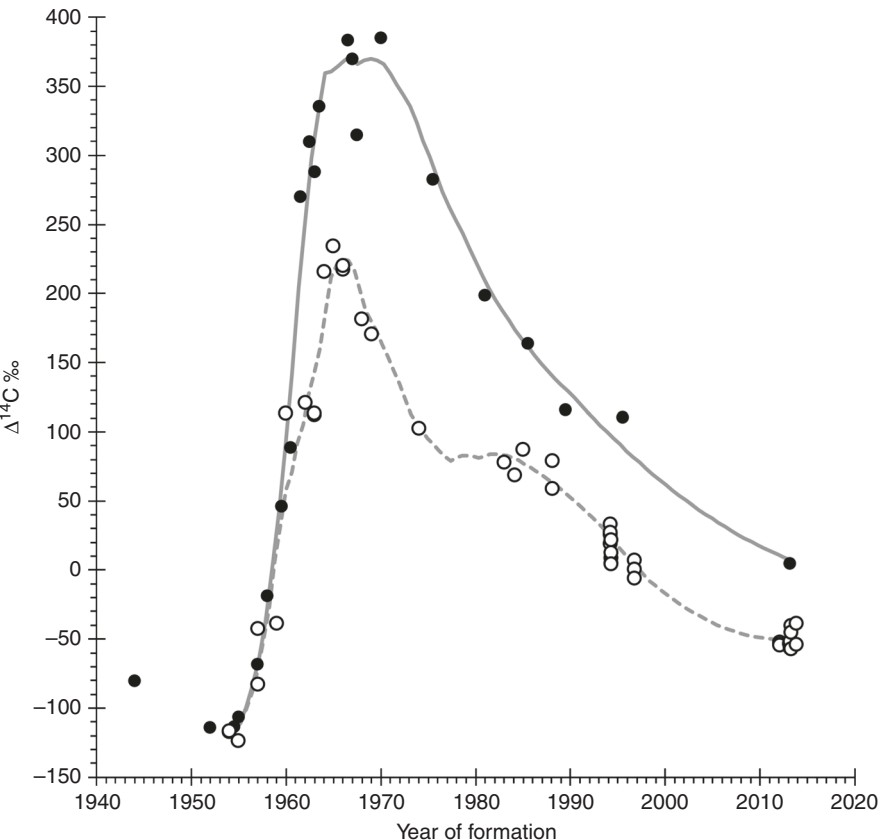

**Fig. 4** Reference freshwater bomb $^{14}$C data. These data provide temporal constraints on the measured values from Bigmouth Buffalo (*Ictiobus cyprinellus*). These records were from a combination of known-age (juvenile fish) and estimated age (core extractions from age-validated adults) to provide the best available bomb-produced $^{14}$C reference record for freshwaters of North America (i.e. salmonids of Arctic lakes (filled circles; *Salvelinus namaycush* and *S. alpinus*)[35] and Freshwater Drum of central North America lakes (open circles; *Aplodinotus grunniens*[46] and U.S. Fish and Wildlife Service unpublished data))

Bigmouth Buffalo of unmarketable commercial size that ranged 2–14 years, with 98% of the individuals between 2 and 6 years old (Fig. 5). Fish from Tenmile Lake (Minnesota River) came from a bowfishing take, and ranged 13–36 years with a dominant 2005 year-class (Fig. 5).

**Growth and reproductive maturity**. Length-at-age estimates from fish used in this study were analyzed for life history parameters using the von Bertalanffy growth function[31]. Six different models were compared in which the parameters for asymptotic length and growth rate were constant or varied with sex (Fig. 6). The global model, in which both parameters varied with sex and $t_0$ unconstrained, was the most parsimonious based on the relative Akaike's Information Criterion adjusted for small sample size[32]. Fixing $t_0 = 0$ produced models that ranked third at best. All but the Minnetaga individuals ($n = 66$) and unsexed Bigmouth Buffalo (skeletons only, $n = 2$) were used in this analysis ($n = 318$). We excluded Lake Minnetaga fish, 98% of which were younger than seven years, because *I. cyprinellus* grow relatively quickly in their first decade (Fig. 6), and these Lake Minnetaga individuals (collected during the fall) had completed an extra growing season compared to all other spring-collected fish in these age classes.

An estimate of reproductive maturity was calculated at the population level from the gonadosomatic index (GSI = gonad mass divided by total fish mass) of individual Bigmouth Buffalo from Artichoke Lake, our only sample in which many Bigmouth Buffalo were taken on a single date in the spring prior to spawning. Totals of 30 females and 14 males were used for this analysis. Sex-specific age at reproductive maturity was determined at the population level following a published method that uses the point at which 50% of the population has mature gonadal tissue[33]. The GSI threshold for which 50% of this population is estimated to reach sexual maturity was approximately the same for males and females (GSI >4%). This corresponds to an age of ~5–6 years for males, and 8–9 years for females. This GSI threshold, although likely appropriate for males, may well be too low for females. This calculation is influenced by the paucity of Bigmouth Buffalo females collected in the range of 6–12 years, and thus our age estimate for female reproductive maturity is likely underestimated for this population. In addition, GSI values are typically 15–25% near asymptotic size for females, while only 5–7% for males, as is true in this case.

**Pigmentation variation with age**. Many Bigmouth Buffalo have unique, long-lasting black or orange markings, and the presence and extent of this pigmentation intensifies with age. In a tagged individual recaptured 9 months later, the position and size of both black and orange spots had not changed (Fig. 7a, b). These color markings are most accentuated in the oldest individuals. Indeed, logistic regression indicated that the presence of black markings increased in likelihood with age ($\chi^2 = 471.425$, $P < 0.0001$). Similarly, orange spots also increased in likelihood with age ($\chi^2 = 415.546$, $P < 0.0001$). Black markings were never found on fish younger than 32 years, yet were present on all individuals older than 45 years (Fig. 7c). Orange spots were present on only two individuals younger than 32 years, and were absent on only four individuals older than 45 years of age. Both black and orange markings vary in position among individuals, but black markings

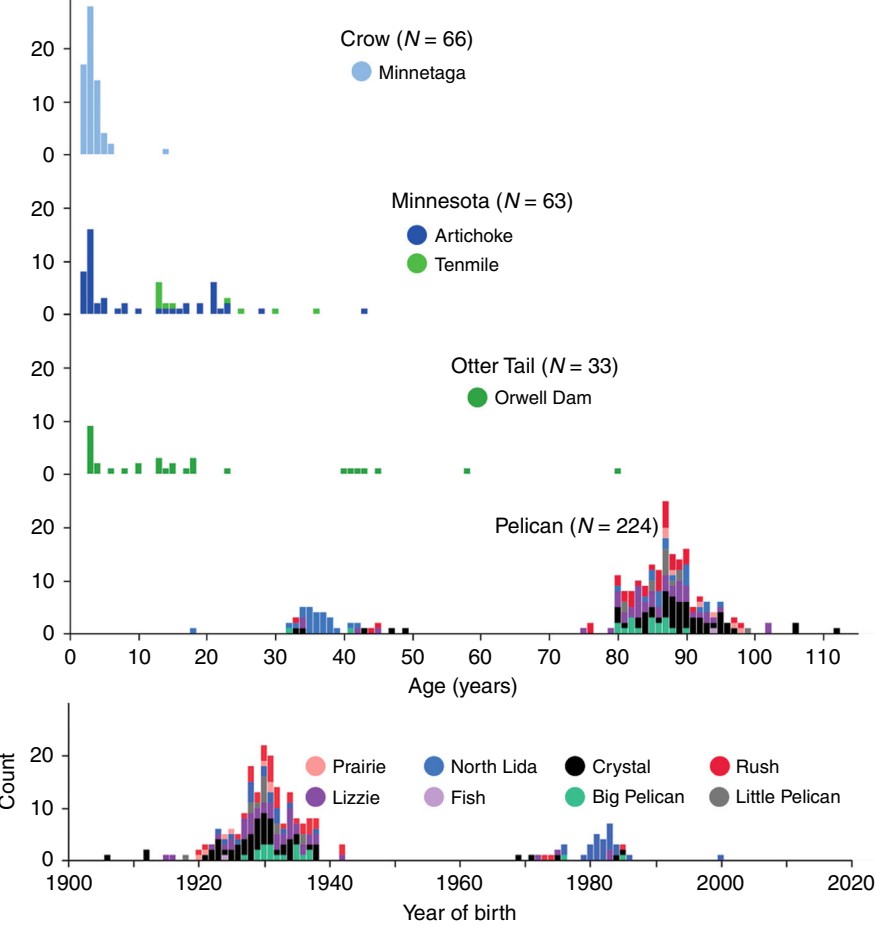

**Fig. 5** Age distribution. Age structure of 2016–2018 sampled Bigmouth Buffalo ($n = 386$) by minor drainage (e.g. Crow) and lake (e.g. Minnetaga). Both age and year class distributions are given for fish from the Pelican River Basin where 83% of the fish are older than 75 years. Although >60% of the individuals bowfished from Tenmile Lake were between 13–15 years, none of the 193 bowfished individuals from the Pelican River Basin was younger than 18, and 81% (156) were over 75 years

usually have a dorsal orientation and orange spots are usually most concentrated on the head.

## Discussion
Taken together, evidence from thin-sectioned otoliths and bomb $^{14}C$ dating revealed that Bigmouth Buffalo can live to 112 years, older than all other reports of maximum age for freshwater teleost fishes by nearly 40 years. To date, the oldest age estimates were from otoliths of Freshwater Drum (*Aplodinotus grunniens*) obtained from archeological sites (maximum reported age of 73 years)[34] and cold-adapted Arctic Lake Trout (*Salvelinus namaycush*; maximum age of 62 years)[35]. With ~12,000 species of freshwater teleost fishes[36,37], the longevity of Bigmouth Buffalo can be considered exceptional. The Family Catostomidae contains at least six other species, representing five of 13 genera, reported to have long lifespans: Quillback (*Carpiodes cyprinus*, 52 years)[38], Razorback Sucker (*Xyrauchen texanus*, 44 years)[39], Cui-ui (*Chasmistes cujus*, 44 years)[40,41], Lost River Sucker (*Deltistes luxatus*, 43 years)[42], June Sucker (*Chasmistes liorus*, 41 years)[43], and Black Buffalo (*Ictiobus niger*, 56 years: a single specimen donated to our research team was 32 years older than the previously reported maximum age)[44]. However, the findings for the Cui-ui and the Lost River Sucker may be underestimates because otoliths were not used. Using otoliths, we show that Bigmouth Buffalo and other catostomids (e.g. Black Buffalo) have life histories that challenge current paradigms. To our knowledge, this is

the first age-validation work done on the buffalofishes (*Ictiobus* spp.), including a first-time application of bomb $^{14}C$ dating to catostomids, and a first-time validation of a freshwater fish lifespan using radial otolith sampling to support ages decades before the bomb $^{14}C$ rise[45]. Thus, Bigmouth Buffalo are now the oldest age-validated freshwater fish.

While bomb $^{14}C$ dating has been widely applied throughout the world, its primary application to fishes has been in the marine environment. Few studies exist that have made thorough assays of the bomb-produced $^{14}C$ signal in the freshwater environment (Figs. 4 and 8), but as with the mixed layer of most of the world oceans, the timing of the rise of $^{14}C$ in freshwater habitats is likely similar across various waterbodies (Fig. 9). Certainly, there are potential complications based on the hydrogeology of the water body under consideration, as exists in the marine environment. However, the utility of the time-specific rise of bomb-produced $^{14}C$ remains an invaluable tool in age validation of freshwater fishes[35,46–48]. In this study of Bigmouth Buffalo, the finding of a valid age-reading protocol to ~60 years, coupled with: the consistency of $^{14}C$ in younger fish with expected $^{14}C$ levels; and radial otolith series for two specimens that push the birth years into the 1920s, clearly supports the validity of age estimates in this study and our conclusion that Bigmouth Buffalo can achieve centenarian longevity.

Over their long lives, Bigmouth Buffalo accrue black and orange spots that correlate with age (Fig. 7). These irregular pigment markings have not been described previously for Bigmouth Buffalo.

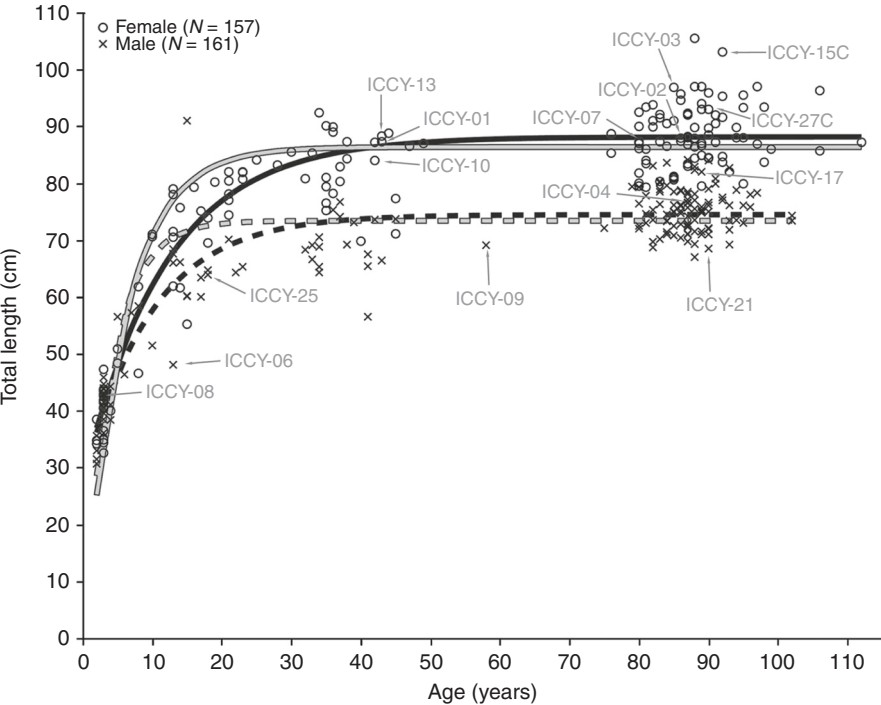

**Fig. 6** Growth in length. Total length (cm) vs. age (years) for sexed Bigmouth Buffalo (excluding Minnetaga) modeled by the highest ranked von Bertalanffy curve (solid black) with different parameters for asymptotic length ($L_\infty$) and growth rate ($k$) for females ($L_\infty = 88.2$, 95% CI [87.2, 89.3], $k = 0.084$ [0.074, 0.096]) compared to males ($L_\infty = 74.5$ [73.5, 75.9], $k = 0.103$ [0.093, 0.114]) (age at 0 length parameter [$t_0$] = −4.4 [−5.7, −3.4]). Both females and males have reached 95% of the asymptotic length by age 30 according to this growth model. Note that $t_0$ is negative due to the absence of 0–1 year old fish in the sample. Fixing $t_0 = 0$ changes the model (gray curves). Bomb $^{14}$C tested Bigmouth Buffalo are labeled with their sample ID (Table 1). Bigmouth Buffalo are taken by bowfishers as small as 30 cm total length in Texas[26]

We hypothesize that black spots accrue from sun exposure over time (melanosis), and that orange spots accrue as a result of diet. Not only do both markings (taken together) provide a consistent, non-lethal means of estimating age (e.g. likelihood of individuals over 75 years old), they also have assisted individual recognition and mark-recapture (Fig. 7). Nonetheless, the utility of these markings has just been realized and their biological function (if any) is unknown. Interestingly, large brown spots were briefly mentioned as a distinguishing feature of old individuals in a different catostomid, the Cui-ui, from Pyramid Lake, Nevada[40].

This revised life history view of Bigmouth Buffalo has implications for management. Dams on rivers are cited as the leading cause of recruitment failure for Bigmouth Buffalo because they restrict access to spawning habitats and can mute the environmental cues thought to initiate spawning behavior[3,5,15,49]. There are four dams along the Pelican River within the eight-lake sampling area (along a 26 km reach of the river), all of which were constructed in 1936–1938 and have been in place for approximately eight decades[50]. Each of these dams restricts upstream movement of fishes. We found the age distribution of the Bigmouth Buffalo populations in the Pelican River Basin lakes to be heavily skewed toward the oldest fish (i.e., 82% of sampled individuals were born prior to 1939, Fig. 5). This is strong indication of a persistent lack of reproductive recruitment since the time of the dam construction.

A further threat to Bigmouth Buffalo populations in Minnesota waterbodies is increased angling pressure since 2010, when regulatory changes permitted angling by night archery with artificial lights[14]. In this form of angling, fish are shot with arrows, catch and release is neither legal nor possible, and there are no bag limits on several endemic taxa including Bigmouth Buffalo and Black Buffalo[14]. Thus, a reevaluation of management decisions concerning Bigmouth Buffalo is required. This new life history

evidence points to a precautionary approach to the conservation of buffalofishes in general, and potentially other catostomids, which currently have little or no harvest regulation. Protecting spawning habitat and older individuals from harvest may be necessary for sustaining populations of species like Bigmouth Buffalo whose life history includes asymptotic growth, delayed maturity, great longevity, and episodic recruitment[51].

In practice, endemic taxa are often ignored if their societal value is not commonly appreciated or has yet to be realized. Addressing such neglect is important in this human-dominated era[52] when ecosystems, literally the life-support system of humankind[53], are destabilized and have lost productivity[54]. For many fishes that are endemic to North America, ecological neglect results from a disregard for the intrinsic value of under-utilized taxa, an under-appreciation for life history diversity, and an inappropriate classification as "rough fish" that portrays low-value to the public. A telling case in this regard is the Bigmouth Buffalo. For centuries this species has been valued as a North American food-fish[11–13], and for decades has served as a direct competitor to several invasive fishes notorious for their deleterious effects on aquatic systems[2,5,16–24]. However, almost all populations of Bigmouth Buffalo in the USA remain unprotected[14] even though they are declining[1]. In contrast, declining populations in Canada have been recognized as a problem, and Bigmouth Buffalo gained Special Concern status in the 1980s[15]. In this study we identify Bigmouth Buffalo as the oldest freshwater teleost, and suggest that urgent conservation measures may be necessary for recovery of old populations with evidence of recruitment failure. Indeed, recent efforts to develop sustainable marine fisheries have emphasized the need to validate lifespans[55], given the threat of longevity overfishing[56]. As was observed for Pallid Sturgeon (Scaphirhynchus albus)[47], it is likely that reproductive and recruitment characteristics associated with a long

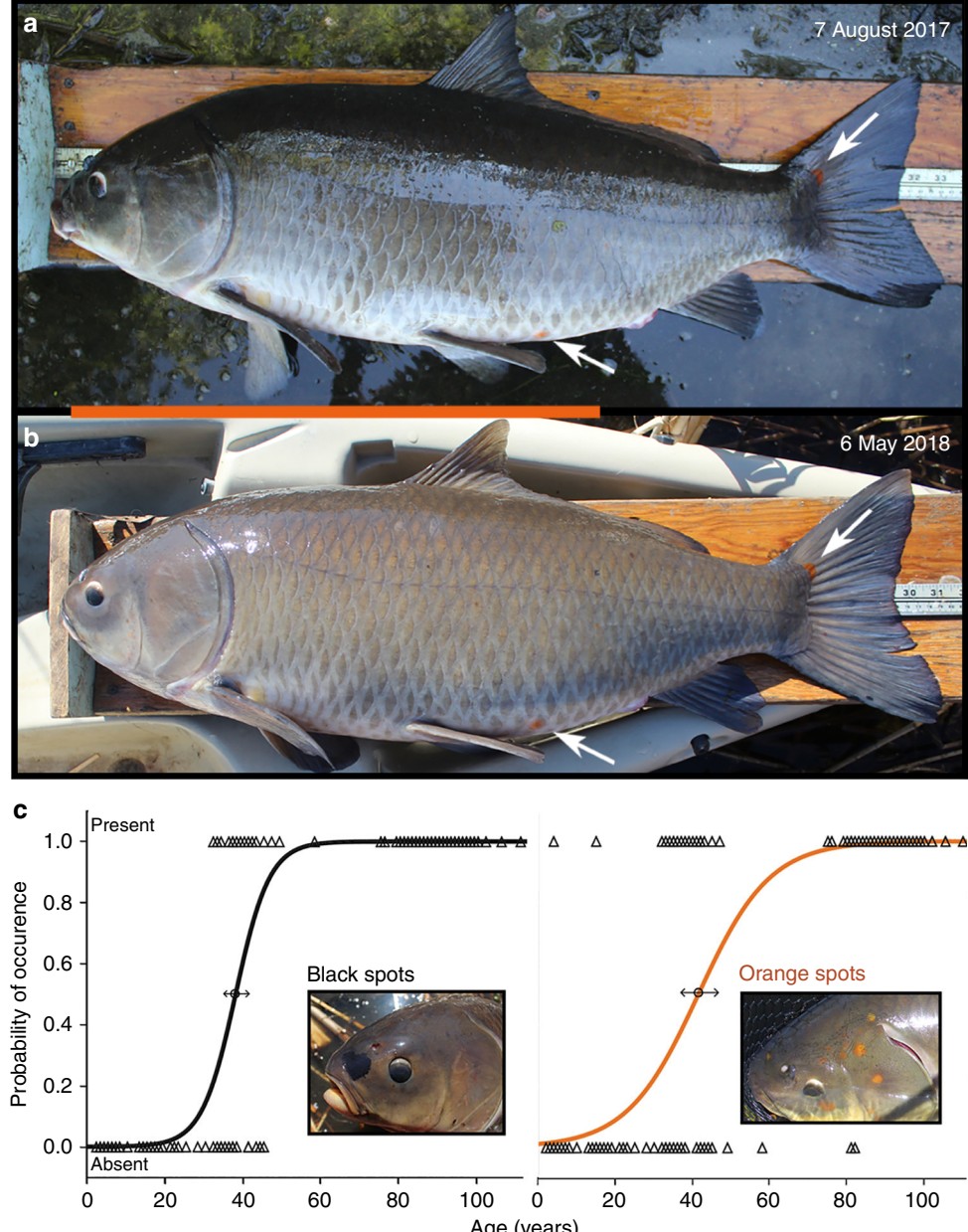

**Fig. 7** Age and pigmentation. **a** An 81.3 cm total length, 9.53 kg female captured in August 2017 had two prominent orange spots (arrows). The fish was tagged with elastomer and released. **b** When recaptured 9 months later she had not grown in total length. Comparing **a** and **b**, these natural orange spots had not changed. Many smaller orange and black spots not obvious in these full-body images also were unchanged. Orange scale bar = 50 cm for both **a** and **b**. **c** The presence of black and orange spots on Bigmouth Buffalo increases in likelihood with age. Data points (triangles) represent presence (1) or absence (0) of these markings on a given fish (n = 384). Inflection points of these logistic regression models are marked with 95% CI. Inset photographs show each type of spot

lifespan may be crucial for population persistence across times of unfavorable environmental conditions common to freshwater habitats. The Bigmouth Buffalo is capable of living and reproducing to ages that more than quadruple all previous estimates. This finding serves as a prime example of discoveries overlooked and management dilemmas that can arise as a consequence of the ecological neglect of under-appreciated species.

## Methods

**Fish collection**. We have treated all animals in accordance with NDSU guidelines on animal care (IACUC protocol A17007).

In the Red River Basin, Bigmouth Buffalo were collected from Otter Tail County, Minnesota, along two tributaries of the main stem of the Red River of the North (henceforth, Red River): the Pelican River and the Otter Tail River. The Pelican River Basin sites included eight lakes: Crystal, Lizzie, Rush, Fish, Pelican, Little Pelican, North Lida, and Prairie. These lakes are along a 26 km reach of the river from which specimens (n = 224) were taken during 2016–2018 via Fyke net, gill net, hook and line, and bowfishing. Otter Tail River Basin individuals (n = 33) came from a single site below Orwell Dam in April of 2018 via hook and line. Specimens were immediately measured to obtain wet mass (±1 g) and total length (±0.1 cm), photographed laterally with a scale bar, and then dissected to obtain gonadal tissue for sex determination and mass (±0.1 g).

In the Mississippi River Basin, Bigmouth Buffalo were collected along two tributaries of the Mississippi River: the Crow River and the Minnesota River. Fish were obtained from a commercial harvest on 4 May 2017 from Artichoke Lake (n = 52) in Big Stone and Swift County, Minnesota (on the Minnesota River); and on 22 Sept 2017 from Lake Minnetaga (n = 66) in Kandiyohi County, Minnesota (on the Crow River). An additional 11 specimens were obtained from a bowfishing take on Tenmile Lake, Otter Tail County, Minnesota (on the Minnesota River) in

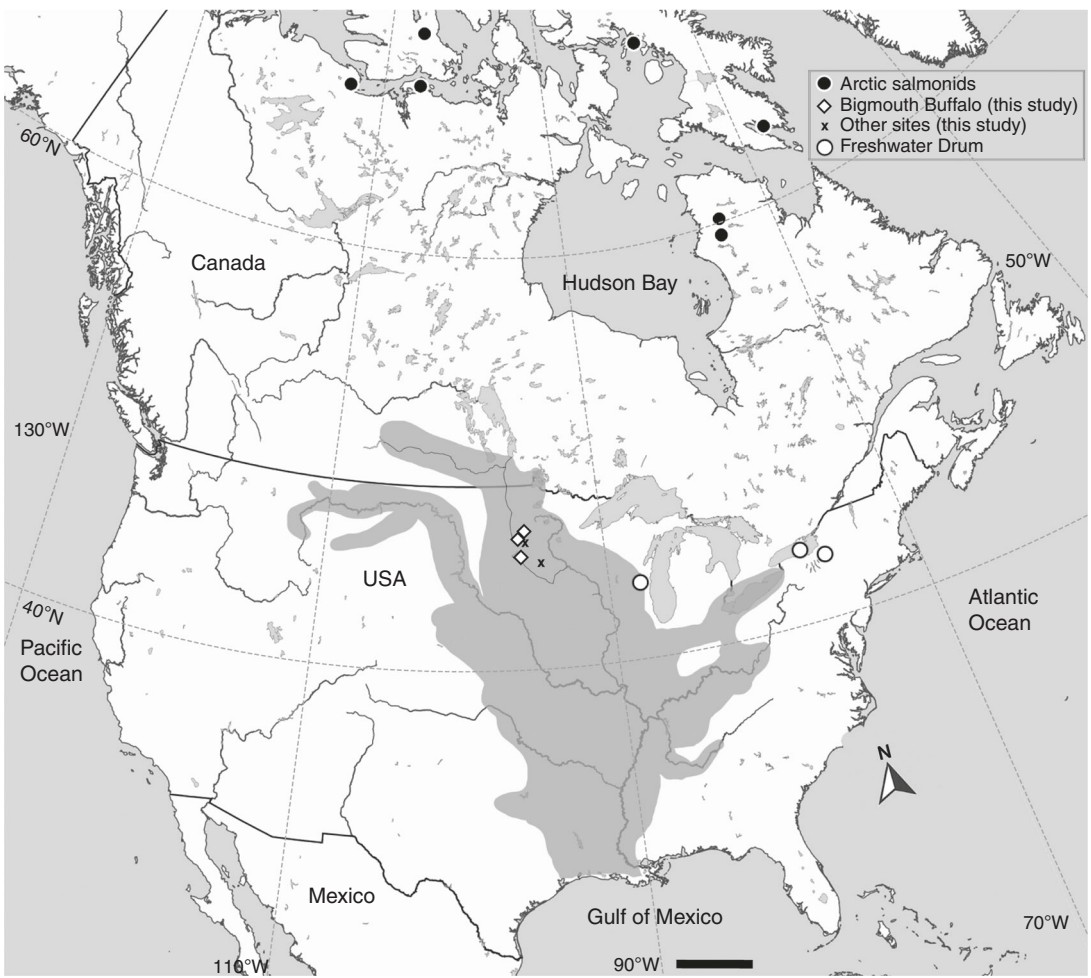

**Fig. 8** Map of freshwater [14]C chronologies in North America. Chronologies have been determined from otoliths of: Arctic salmonids[35]; Freshwater Drum of Lake Winnebago (western white circle)[45] and Lake Ontario and Lake Oneida (eastern most white circles; U.S. Fish and Wildlife Service unpublished data); and 3) Bigmouth Buffalo (*Ictiobus cyprinellus*) of Minnesota (diamonds; present study). Bigmouth Buffalo were also taken from the points marked with an "X", but these were not analyzed for radiocarbon. The dark-gray shaded area within the USA and Canada represents the endemic range of Bigmouth Buffalo[5,62]. Scale bar = 400 km

May of 2018. For Artichoke Lake fish, measurements, photographs, and sex determination were obtained after fish had been frozen and thawed. For Lake Minnetaga and Tenmile Lake, fish data were obtained as previously described for the Red River Basin specimens, except that Lake Minnetaga specimens were dissected for sex determination after being frozen and thawed.

**Otolith preparation and age analysis**. Otoliths were removed from fish by first exposing the ventral surface of the cranium, through the otic bullae under the operculum. At least one otolith was obtained from every fish dissected ($n = 386$) and in most cases (77%) the complete set of six otoliths (asterisci, sagittae, and lapilli) was collected. Following extraction, the otoliths were gently removed from the labyrinth organ with forceps and placed in 1.5 ml plastic microvials pre-filled with water to prevent any residual tissue or fluid from drying to the surfaces. All collected otoliths were rinsed and submersed in distilled water to photograph the whole otolith set at 10X with an Olympus® SZH10 dissecting microscope using transmitted light in dark-field mode. The orientation of the nuclear transect to be thin-sectioned from the whole otolith was determined from these images. Otoliths were then dried for 30 min at 55 °C and lapilli were weighed (±0.001 g) using a CAHN Electrobalance®. Only the lapilli were weighed because they produce the most reliable weight measures. Of all Bigmouth Buffalo otoliths, lapilli are the largest, least fragile, and least likely to hold residual endolymph. Sagittae are the smallest otoliths in Bigmouth Buffalo and fracture easily, while asterisci have grooves that are difficult to thoroughly clean of non-otolith material (both factors that led to unreliable weight measures).

Weighed otoliths were embedded in ACE® quick-setting epoxy within 1.5 cm³ compartments (lined with petroleum jelly) in a plastic tray. After the epoxy hardened, the epoxy block was placed in a Buehler IsoMet™ 1000 low-speed saw equipped with diamond-embedded thin-sectioning blades to obtain 300–500 µm

sections via the wafer method. A total of 557 otoliths (315 asterisci, 241 lapilli, and 1 sagittal) were thin-sectioned to obtain age estimates for these 386 Bigmouth Buffalo. Sagittae are the most difficult otoliths to section in Bigmouth Buffalo, and thus were rarely used. Sections of asterisci and lapilli from the same individual produce essentially the same age estimate for the entire range of ages (Figs. 1 and 2), proving that either structure can be used. For 165 individuals, only asterisci were thin sectioned, and for 114 specimens only lapilli. The remaining specimens had both asterisci and lapilli ($n = 106$), or lapilli, asterisci, and sagittae ($n = 1$) thin sectioned to provide comparison opportunities within individual fish. In addition, both asterisci from a single specimen were sectioned for 25 individuals, and both lapilli for 11 individuals. Although many sections were taken, a small portion (~15%) were too fractured or structurally polymorphic to be readable. Nonetheless, at least one readable section from every Bigmouth Buffalo in this study was obtained.

Thin sections of the otoliths were mounted on a glass microscope slide and immersed in mineral oil to enhance visibility and photographed at 75× under a compound microscope using transmitted light. Multiple images per thin section were required to provide a composite image of the whole otolith section at this magnification. Images were stitched together using Adobe Photoshop software to create the high-resolution composite image of the whole thin section. The composite images were then examined for annuli that could be quantified and were digitally marked (Fig. 1).

The best otolith sections were assigned ages by multiple readers, with consensus readings used to determine the final age assigned to each specimen. First, a primary and secondary reader independently marked annuli on duplicate images of the thin section. Discrepant annuli counts between the primary and secondary reader were identified using a minimum criterion of 1 year per decade of age. For example, reader counts for individuals scored 0–9 years of age were deemed discrepant if the primary and secondary reader scores differed by more than ±1 annulus count.

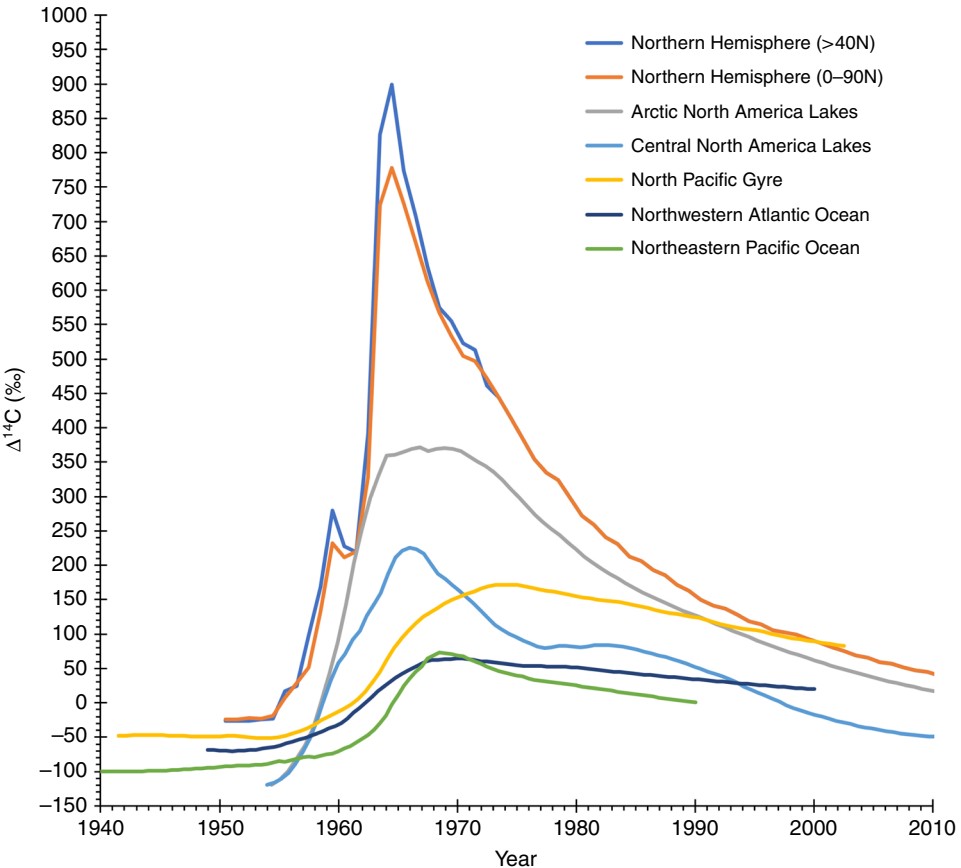

**Fig. 9** Various bomb-produced $^{14}$C records. Note the general similarities and differences in each environment of the Northern Hemisphere. The most rapid increase is atmospheric due to thermonuclear testing in this environment for which two $^{14}$C data sets were applicable to North America (above 40° north and an average of various records across all northern latitudes[63]). Arctic and Central lakes of North America also exhibit a rapid $^{14}$C rise due to a close hydrologic connection via precipitation. The marine environment is exemplified by a coral record from the North Pacific Gyre (*Porites* sp. of Kure Atoll)[64] and two fish otolith records from the upwelled environment of the northeastern Pacific (Yelloweye Rockfish, *Sebastes ruberrimus*)[61] and the mixed layer of the northwestern Atlantic (various species)[35]

This approach was used for individuals scored up to 110–119 years (deemed discrepant if the primary and secondary reader scores differed by more than ±12 annulus counts). If reader scores fell into separate decades, the younger age group criterion was used. Images of otoliths identified as discrepant based on these criteria were then either independently analyzed by a third reader ($n = 29$), or another otolith section(s) already available from the same fish was aged by both primary and secondary readers. If consensus scores were still not obtained between readers, then yet another otolith was thin-sectioned from that specimen and again scored independently by the primary and secondary reader, at which point all age estimates were resolved. Otoliths for which annuli counts were not identical between readers, but not identified as discrepant (e.g., scored 12 by the primary and 13 by the secondary), a final determination was made by the primary reader. The overall between-reader precision (primary and secondary) was a coefficient of variation (CV) of ~5.6%. This precision varied with age and was highest in the youngest group of fish, as expected. For individuals across each of the 12 decadal age groups in this study (from 0–9, to 110–119 years) the precision was CV ~10.4, 5.7, 4.0, 4.5, 4.5, 3.6, NA, 3.3, 2.9, 3.4, and 2.7, and 3.9% respectively.

**Bomb radiocarbon dating**. We selected for radiocarbon analysis 15 lapillus otoliths from Bigmouth Buffalo previously aged via a thin-sectioned asteriscus or lapillus (or both) annulus count(s). These fish spanned the range of chronological dates required for this type of age validation work (Table 1). Typically, a selection of birth years that range from the pre-bomb period (earlier than ~1955) to the post-peak decline period (more recent than the 1970s) is used to trace the bomb-produced $^{14}$C signal through the lifespan of the species, and to potentially provide diagnostic ages from birth years associated with the rapid rise of $^{14}$C in the 1950s and 1960s. For dating, we chose a lapillus to the matching, thin-sectioned asteriscus or lapillus (or both) used for age determination (Table 1), because lapilli are the largest otoliths by mass for Bigmouth Buffalo and thus were most likely to provide a sufficient amount of calcium carbonate for $^{14}$C analyses. The 15 lapillus otoliths selected for bomb $^{14}$C dating were sectioned in a similar manner to the previously

described thin sectioning, except that they were serially sectioned using a single blade to a thickness of ~1 mm. We selected a section that contained the desired core (the first 1–2 years of growth), with a planar orientation normal to the growth layer structure, such that the growth layers were not tilted and the micromilled material would include only the targeted growth years. A section thickness of 1 mm was necessary to provide greater material depth for micromilling and sufficient mass for $^{14}$C analysis.

Otoliths were micromilled using a New Wave Research micromilling machine to a depth of ~600–800 μm providing ~0.5–1.3 mg of carbonate per sample (Table 1). A total of 15 specimens spanning the bomb $^{14}$C chronology were milled for the core region of the otolith, representing the first 1–2 years of growth, and for 13 of these, only the core was extracted. From the two additional individuals (both estimated to have hatched prior to atmospheric nuclear testing in the 1950s and 1960s), multiple samples were extracted per otolith in a radial pattern that began after the core extraction and led into more recent years of formation (Table 1). The goal was to detect the location in the otolith section (year or years of formation) where the time-specific rise of bomb-produced $^{14}$C occurred (~1955). This approach can validate age estimates exceeding the minimum maximum age indicated by pre bomb radiocarbon levels in the otolith core. The radial extractions were assigned a mean year of formation by overlaying the annulus structure (an image from the aged lapillus section) on an image of the path extracted by the micromill.

We submitted 28 extracted otolith samples as carbonate to the National Ocean Sciences Accelerator Mass Spectrometry Facility (NOSAMS), Woods Hole Oceanographic Institution in Woods Hole, Massachusetts, for $^{14}$C analysis. Radiocarbon measurements were reported by NOSAMS as Fraction Modern (the measured deviation of the $^{14}$C/$^{12}$C ratio from Modern). Modern is defined as 95% of the $^{14}$C concentration of the National Bureau of Standards Oxalic Acid I standard (SRM 4990B) normalized to δ$^{13}$C VPDB (–19‰) in 1950 AD (VPDB = Vienna Pee Dee Belemnite standard)[57]. Radiocarbon results were corrected for isotopic fractionation using a value measured concurrently during the accelerator mass spectrometry analysis, and these data are reported here as F$^{14}$C. These values

were date corrected based on the estimated year of formation and are reported[58] as $\Delta^{14}C$. Stable isotope $\delta^{13}C$ measurements were made on a split of $CO_2$ taken from the $CO_2$ generated during acid hydrolysis. These values are robust and can be used to infer carbon sources in the formation of the otolith carbonate.

Measured $\Delta^{14}C$ values were used to determine the validity of age estimates by comparing the purported year of formation (birth year), calculated from the collection year and estimated age relative to regional $\Delta^{14}C$ references (Figs. 4, 8, and 9). Temporal alignment of the measured $\Delta^{14}C$ values from otolith material with regional $\Delta^{14}C$ reference records from otoliths of other freshwater fishes provided an independent basis for determining fish age, and for evaluating our age reading protocol for Bigmouth Buffalo based on otolith annulus counts. The only thorough $^{14}C$ reference records available for the freshwater environment of North America were from arctic lakes and mid-continent lakes near the Great Lakes, because very little work has been done in this regard (Fig. 8).

### Freshwater radiocarbon references for North America.
Overall, bomb radiocarbon dating is considered one of the best methods of age validating long-lived fishes[30]. The radiocarbon ($^{14}C$) data used as reference material to validate the age and longevity of Bigmouth Buffalo (*Ictiobus cyprinellus*) in this study were from a series of rare freshwater sources that used otoliths of two fish species from widely separated regions of North America (Fig. 8). These bomb $^{14}C$ records were from otoliths of either known-age (juvenile fish) or aged adults (otolith cores) from: salmonids of Arctic lakes (*Salvelinus namaycush* and *S. alpinus*)[35], and Freshwater Drum of central North America lakes (*Aplodinotus grunniens*[45] and U.S. Fish and Wildlife Service - Northeast Fishery Center, Lamar, Pennsylvania, unpublished data). These data sets were fitted with a Loess curve (spline interpolation smoothing parameter = 0.4, two-parameter polynomial; SigmaPlot v.11.2) to describe the central tendency of each time series (Fig. 4). A caveat of the curve fitting is that one Freshwater Drum specimen from Lake Ontario (2012) was elevated relative to all others from Oneida Lake in 2012–2014 and was considered more likely to be similar to the Arctic references due to hydrography of the Laurentian Basin. This $^{14}C$ value may have been elevated due to increased atmospheric exposure from greater water mass residence times in the Great Lakes (relative to Oneida Lake) along with other catchment factors associated with the delivery of terrestrial carbon sources that can be $^{14}C$-enriched[59]. The bomb-produced changes in freshwater $^{14}C$ for North America may begin with what appears to be variable $^{14}C$ levels in the pre-bomb period ($\Delta^{14}C$ ranged from approximately −80‰ to −125‰ before 1955) and become coincident as the sharp bomb-produced $^{14}C$ rise begins near 1955 (Fig. 4). At mid-rise, near 1960, the regional records (Arctic vs. Central North America lakes) start to diverge and then exhibit differences in peak amplitude and subsequent decline. A separation of the records is maintained through the decline period of the 1970s to the 2010s, but the signal appears to dovetail toward most recent years, provided the elevated specimen from Lake Ontario is an accurate reflection of regional variability. Regardless of the potential for minor variability in $^{14}C$ levels, the $^{14}C$ rise due to atmospheric testing provides a valid marker that can be used to determine the validity of age estimates, with further support from the generally consistent pattern of the overall rise and fall of bomb-produced $^{14}C$.

Very little work has been done with determining the full bomb-produced $^{14}C$ signal in freshwater environments — most has been within the marine environment (usually in the mixed layer using various forms of biogenic carbonate). There are differences between the bomb-produced $^{14}C$ signals in these environments, primarily because of the way $^{14}CO_2$ from nuclear testing enters the hydrologic system. While input of the bomb $^{14}C$ signal to the ocean system relies mostly on air-sea diffusion at the sea surface, the freshwater environment has a more direct advection of bomb $^{14}C$ from the atmosphere to rivers and lakes via precipitation. Hence, the hydrology of the freshwater environment leads to a more synchronous link to $^{14}C$ changes in the atmosphere and exhibits a more rapid $^{14}C$ rise than the marine environment (Fig. 9). The $^{14}C$ peaks expressed for the Arctic and central North America lakes may be artificially muted because actual peak dates may not have been sampled[35]. Nonetheless, the marine bomb-produced $^{14}C$ signal is usually attenuated and phase lagged relative to both freshwater and atmospheric $^{14}C$ records (Fig. 9). The exceptions are either, close-in fallout that generated a strong regional $^{14}C$ signal in the marine environment[60], or places where there are $^{14}C$-depleted sources from either unique hydrogeology (karst topography; AH Andrews, pers. observation) or upwelled waters of the deep-sea[61]. For the existing freshwater $^{14}C$ records it is the temporal similarities, despite differences in amplitude, that indicate tracing the bomb-produced $^{14}C$ signal in other freshwater environments of North America (e.g. river basins of Minnesota for Bigmouth Buffalo). These temporal constraints on otolith $^{14}C$ measurements can be used to validate age estimates.

In some cases, otolith core material cannot be used as a strong record of support for determining the age of other organisms because of reasoning circularity — a fish of unknown age that was age-validated from a reference $^{14}C$ record should not be in turn used as a reference to validate the age of other otolith measurements of unknown age. However, this is avoidable when otolith annuli are very well defined and there is little or nothing else to refer to as a regional $^{14}C$ reference. If the temporal nature of the nearest regional $^{14}C$ signal is a match with the otolith material's signal (position in time based on annulus counts from the otolith), then an assumption can be made that adults of the species can provide a

bomb-produced $^{14}C$ timeline where none existed. Hence, this is the case for both the Arctic salmonids and Freshwater Drum used as a reference in the current study on Bigmouth Buffalo. Known age juvenile fish and cored adults with well-defined otolith annuli produced strong evidence of the regional $^{14}C$ signal of freshwater environments in North America. This data provides a strong basis for validating other freshwater fishes in this region (e.g. Bigmouth Buffalo). These are the most complete records for this environment. The only other records for freshwater environments of North America come from Lake Sturgeon (*Acipenser fulvescens*)[46] and Pallid Sturgeon *Scaphirhynchus albus*)[47], but these $^{14}C$ records are not as complete.

### Mark recapture.
Bigmouth Buffalo from Big Pelican and Little Pelican Lakes (Otter Tail County, Minnesota) were captured during 2011–2018 by hook-and-line. Captured individuals were photographed, measured (total length and wet mass, as described previously), sexed (based on visual examination of the uro-genital opening, presence of tubercles, or expression of gametes (or combination thereof)), tagged at initial capture using either safety pins or Visible Implant Elastomer tags (Northwest Marine Technology, Inc.), and released in good condition.

### Statistics and reproducibility.
We used JMP Pro Statistical Discovery™ Software (Version 13.0, SAS Institute, Inc. 2014) for statistical analysis and graphical output. SigmaPlot (Version 11.2) was used to render smoothed curve fits (Loess function, spline interpolation smoothing parameter = 0.4, two-parameter polynomial) to the regional $^{14}C$ reference data (Figs. 3, 4).

### Reporting summary.
Further information on research design is available in the Nature Research Reporting Summary linked to this article.

## Data availability
The data that support the findings of this study are available from the corresponding author upon reasonable request.

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

## Acknowledgements

We thank Kurt Lackmann, Derek Knuti, Matt Guck, Jeffrey Lackmann, Derek Sauer, Neil Lelm, Claire Wiseman, Reed Jacobson, Kathy Lackmann, Deb Lackmann, Mollie Lackmann, Alex Huseby, Clayton Lewandowski, Kui Hu, Ethan Rasset, Jason George, Mike Sands, and Richard Venero for assistance in the lab or field; Scott Payne and Jayma Moore of the NDSU Electron Microscopy Lab for use of their Isomet saw; Dave Majkrzak for donation of an Isomet saw; NOAA Fisheries, Pacific Islands Fisheries Science Center for use of the micromilling facility; Greg Jacobs and Jonah Withers of the US Fish and Wildlife Service for use of freshwater drum $^{14}$C reference data, and Dr. Ann McNichol of the National Ocean Sciences Accelerator Mass Spectrometry Facility at Woods Hole Oceanographic Institution; Funding: Woods Hole Oceanographic Institution, National Ocean Sciences Accelerator Mass Spectrometry, Research Initiative: National Science Foundation Cooperative Agreement number, OCE-1239667; Department of Biological Sciences, Environmental and Conservation Sciences Program, NDSU; We thank the Vice President of Research and Creative Activity at NDSU, Jane Schuh, for facilitating additional radiocarbon analysis with an anonymous donor; North Dakota Water

Resources Research Institute. This is University of Hawaii, School of Ocean and Earth Science and Technology publication number 10701.

## Author contributions

A.R.L. conceived the study and wrote the first draft of the manuscript; A.R.L., A.H.A. and M.E.C. contributed to data collection, analyses, and manuscript writing; E.B.L. contributed to data collection and analyses; and M.G.B. contributed to analyses and writing

## Additional information

**Competing interests:** The authors declare no competing interests.

