## [Peer Review File · Communications Biology]

REVIEWERS' COMMENTS:

Reviewer #1 (Remarks to the Author):

Review of "Centenarian longevity of Bigmouth Buffalo" by Lackmann et al
Dear Authors,

It has been a pleasure to read your manuscript, the language is fluent and the paper is well organised. The major claim of the paper is that the age of the Bigmouth Buffalo to reach 112 years using the annual layers of otoliths and 14C analysis. I generally find this statement justified by the data and methods presented. Further, I do believe that the present paper will have influence on management legislation of the Bigmouth Buffalo in USA.

The authors have done a great work on freshwater reference curves. This is really an important tool for interpreting the presented data. These data are extremely useful for other researchers too. However, I think the authors view a too simplistic on explaining the different DIC sources to freshwater system. I would like to see this to be further elaborated. Also, I think that these curve are denoted local / site specific or regional as they will be different for different freshwater systems. This is important to stress to ensure correct use by other researchers in the future. I don't see that this will have any influence on the age estimate of the Bigmouth Buffalo.

Line 69 ", so and" something missing here?

Line 85 "With the exception of Pennsylvania.." maybe replace with "In Pennsylvania ..."

Line 101 "estimated the age"

Line 118 Please explain expected 14C levels / reference records – might not be clear to the reader what this at this point

Line 127 For reference records shown this surely appears to be the case, however, as also stated elsewhere freshwater reference records are rare – so can you be sure this is always the case for all freshwater systems? (I think not – I believe these may be very complex and site-specific due to geology, geochemistry, hydrology and so forth)

Line 128 I think 'expectation' is an overstatement. Are these freshwater reference curves universal to the extend that you may expect other data to follow the curve? I would slightly weaken the statement.

Line 135 please explain what you mean with 'expected'

Line 142 this is more comment to statement above. This exactly shows that freshwater systems are complex and variable, and hence that reference curves for freshwater systems must be considered local.

Line 194 I don't think you can state that 14C bomb dating can reveal ages older than 1960 – 50. I am a little ambivalent on what to think here. Clearly 14C bomb dating is important in order to make this conclusion, however, what you really do is to verify your otolith annual counts using the bomb rise. On its own 14C cannot be used to infer the old age (unless you construct a model e.g. Nielsen et al, Science, 2016). Maybe attach a small explanatory paragraph after the sentence

Line 214 Would change to "timing of the rise" is similar. As correctly explained in line 217. It may read as curves are similar.

Fig 4 Out of curiosity: Do you have any ideas on why the data provides a negative t0? Is it due to some unknown systematic error or is just noise?

Line 427 maybe add a reference for extraneous material

Line 506 Shouldn't this the ref 59 (not 58). How is the Delta14C calculated?

Fig 4 what does Hem (mean) denote – and how is it calculated? Perhaps add curve for north Atlantic too.

Fig 2ext please add a legend

Supplementary

Line 19. Well couldn't this be due to catchment processes or precipitation delivering a 'terrestrial' DIC/DOC signal to the lake? I don't think residence time alone explains (or at least other possibilities should be mentioned) this. Numerous catchment processes may deliver enriched ^{14}C to the lake and rivers either directly as DIC or as DOC which may be partly oxidised to DIC.

See e.g.

Keaveney, E. M., Reimer, P. J. & Foy, R. H., 2015, In : Radiocarbon. 57, 3, p. 407-423 17 p.

Keaveney, E. M., Reimer, P. J. & Foy, R. H., 2015, In : Radiocarbon Journal. 57, 3, p. 425-438 14 p.

Line 28 From fig 3 and 4 I think it would be unwise to use fall for any dating attempts.

Line 34 DIC in oceans surface is a mixture of air-sea gas exchange between surface and atmosphere as well as carbon exchange between surface and deep ocean. Look up references for the Marine calibration. See e.g. Stuiver, M. and T. F. Braziunas (1993). "Modeling atmospheric ^{14}C influences and ^{14}C Ages of Marine Samples to 10,000 BC." Radiocarbon 35: 137-189.

Line 35 surely precipitation plays a role, however, what about land-use, catchment geochemistry etc These are important in delivering carbon to freshwater systems. On top of this carbon cycling and recycling within lakes may further be significant for some lake types.

Line 56 yes – I like that term 'regional' and please add the data from 47 and 48 to figure 3 and 4

Reviewer #2 (Remarks to the Author):

I have completed my review of the manuscript "Centenarian longevity for Bigmouth Buffalo *Ictiobus cyprinellus*" by Alec R. Lackmann, Allen H. Andrews, Malcolm G. Butler, Ewelina S. Bielak-Lackmann and Mark E. Clark.

I found this manuscript to be well-written and very interesting. The experimental design is appropriate, yielding clear and important results, that can help fisheries scientists to make better informed conservation decisions.

I've added a few very minor editorial suggestions in WORD Track Changes. The figures are well done and serve to enhance the paper. The paper is suitable for publication in Communications Biology, and I commend the authors for their work. You may identify me to the authors.

Response to referees and editors in point-by-point manner (all of these line numbers refer to our revised manuscript with ‘track changes’ expanded)

Lines 1-3: Title modified (advised by editors in AIP document)

Lines 13-14: phone number deleted (advised by editors in AIP document)

Lines 16-31: Abstract revised (advised by editors in AIP document)

Lines 33-35: Keywords and short title deleted (advised by editors in AIP document)

Lines 50, 52, 54, 61: BMB acronym replaced with common or species name of species (advised by editors in AIP document)

Line 61: word “significant” replaced with “important” (advised by editors in AIP document)

Lines 63-65: minor technical word changes (buffalo fish should be one word, i.e. buffalofish)

Line 67: BMB acronym replaced with common or species name of species (advised by editors in AIP document)

Lines 69-73: minor wording changes; BMB acronym replaced with common or species name of species (advised by editors in AIP document)

Line 76: BMB acronym replaced with common or species name of species (advised by editors in AIP document)

Line 78-80: minor wording changes in response to Reviewers 1 and 2:

Reviewer 1’s comment: *[formerly] Line 69 “, so and” something missing here?*

(Reviewer 2 simply made an editorial revision);

BMB acronym replaced with common or species name of species (advised by editors in AIP document)

Lines 85-86: BMB acronym replaced with common or species name of species (advised by editors in AIP document)

Lines 91-97: minor wording changes in response to Reviewer 1's comment: *[formerly] Line 85 "With the exception of Pennsylvania.."* maybe replace with *"In Pennsylvania ..."* (see track changes document for more details);
BMB acronym replaced with common or species name of species (advised by editors in AIP document)

Lines 101-106: minor wording changes to the present tense and BMB acronym replaced (advised by editors in AIP document);

Line 113: minor wording changes in response to Reviewer 1's comment: *[formerly] Line 101 "estimated the age"*; BMB acronym replaced (advised by editors in AIP document)

Line 119: PRB acronym added so at least 5 of these acronyms are throughout the text (advised by editors in AIP document)

Line 122: BMB acronym replaced (advised by editors in AIP document)

Lines 126-127: minor wording changes to reflect that we no longer have SI (advised by editors in AIP document)

Lines 131-133: revision in response to Reviewer 1's comment: *[formerly] Line 118 Please explain expected 14C levels / reference records – might not be clear to the reader what this at this point*

Line 136: minor wording change to reflect correct figure order after we no longer have SI (advised by editors in AIP document)

Line 139: corrected minor error, and minor wording change to reflect correct figure order after we no longer have SI (advised by editors in AIP document)

Line 142: revision in response to Reviewer 1's comment: *[formerly] Line 127 For reference records shown this surely appears to be the case, however, as also stated elsewhere freshwater reference records are rare – so can you be sure this is always the case for all freshwater systems? (I think not – I believe these may be very complex and site-specific due to geology, geochemistry, hydrology and so forth)*

Line 143: revision in response to Reviewer 1's comment: *[formerly] Line 128 I think 'expectation' is an overstatement. Are these freshwater reference curves universal to the extent that you may expect other data to follow the curve? I would slightly weaken the statement.*

Line 147: BMB acronym replaced (advised by editors in AIP document)

Lines 149-154: minor wording change to reflect that we no longer have SI (advised by editors in AIP document); revision in response to Reviewer 1's comment: *[formerly] Line 135 please explain what you mean with 'expected'*

Lines 158-159: minor wording change to reflect that we no longer have SI (advised by editors in AIP document); minor wording change in response to Reviewer 1's comment: *[formerly] Line 142 this is more comment to statement above. This exactly shows that freshwater systems are complex and variable, and hence that reference curves for freshwater systems must be considered local. See 'track changes' document for more detailed response in the comment section.*

Line 163: acronym edits (advised by editors in AIP document)

Line 166: minor wording change to reflect that we no longer have SI; and acronym edit (advised by editors in AIP document)

Line 168-173: minor wording changes to reflect that we no longer have SI; and acronym edits (advised by editors in AIP document)

Lines 178, 181, 183-184, 187-188: minor edits because we no longer have SI; acronym edits (advised by editors in AIP document)

Lines 190-192: expanded on methodological procedure instead of just citing a reference: (advised by editors in AIP document)

Lines 195-196: BMB acronym replaced (advised by editors in AIP document); minor wording change

Lines 201, 203, 207, 209: acronym edit; minor wording changes to reflect that we no longer have SI (advised by editors in AIP document)

Lines 214-216: acronym edit (advised by editors in AIP document); minor wording changes in response to Reviewer 1's comment: *[formerly] Line 194 I don't think you can state that 14C bomb dating can reveal ages older than 1960 – 50. I am a little ambivalent on what to think here. Clearly 14C bomb dating is important in order to make this conclusion, however, what you really do is to verify your otolith annual counts using the bomb rise. On its own 14C cannot be used to infer the old age (unless you construct a model e.g. Nielsen et al, Science, 2016). Maybe attach a small explanatory paragraph after the sentence. Please see 'track changes' document for detailed response in the comment section*

Line 219: acronym edit (advised by editors in AIP document)

Lines 226-230: wording changes as result of removing 'data not shown' statement; acronym edit (advised by editors in AIP document); "buffalo fishes" revised to one word, as it should be, technically

Lines 232-233: Additional point added to make concluding statements of paragraph clear.

Lines 236-238: minor wording changes to reflect that we no longer have SI (advised by editors in AIP document); minor wording change in response to Reviewer 1's comment: *[formerly] Line 214 Would change to "timing of the rise" is similar. As correctly explained in line 217. It may read as curves are similar.*

Lines 241, 244, 246-248, 251, 255-256: acronym edits; minor wording changes to reflect that we no longer have SI (advised by editors in AIP document)

Lines 260-262: minor wording change (done on our own); acronym edit, and minor wording changes to reflect that we no longer have SI (advised by editors in AIP document)

Lines 264, 267-269, 272, 280, 282-285: acronym edits (advised by editors in AIP document); and minor wording change (done on our own) to make more descriptive; "buffalo fishes" revised to one word, as it should be, technically

Line 287: minor wording change

Line 292, 297, 302, 306: acronym edits (advised by editors in AIP document)

Lines 328-329: acronym edits (advised by editors in AIP document), and wording change in response to Reviewer 1's comment: *[formerly] Line 427 maybe add a reference for extraneous material.* Please see 'track changes' document for detailed response in the comment section

Lines 336-337, 339, 345, 373, 381: acronym edits, and minor wording change to reflect that we no longer have SI (advised by editors in AIP document)

Line 389: corrected minor error

Line 405: wording change (as Reviewer 2's editorial change advised)

Line 408: revision in response to Reviewer 1's comment: *[formerly] Line 506 Shouldn't this be the ref 59 (not 58). How is the Delta14C calculated?* Please see 'track changes' document for detailed response in the comment section

Lines 414, 417, 419-420, 424-426, 431: acronym edits; minor wording changes to reflect that we no longer have SI (advised by editors in AIP document)

Line 433: wording change (as Reviewer 2's editorial change advised)

Lines 436-437: wording change (as Reviewer 2's editorial change advised); revision and citation addition in response to Reviewer 1's comment: *[formerly] Line 19 [in SI]. Well couldn't this be due to catchment processes or precipitation delivering a 'terrestrial' DIC/DOC signal to the lake? I don't think residence time alone explains (or at least other possibilities should be mentioned) this. Numerous catchment processes may deliver enriched 14C to the lake and rivers either directly as DIC or as DOC which may be partly oxidised to DIC.*

See e.g.

Keaveney, E. M., Reimer, P. J. & Foy, R. H., 2015, In : Radiocarbon. 57, 3, p. 407-423 17 p.

Keaveney, E. M., Reimer, P. J. & Foy, R. H., 2015, In : Radiocarbon Journal. 57, 3, p. 425-438 14 p.

Line 440: minor wording changes to reflect that we no longer have SI (advised by editors in AIP document)

Lines 444-448: revision in response to Reviewer 1's comment: *[formerly] Line 28 [of SI] From fig 3 and 4 I think it would be unwise to use fall for any dating attempts.*

Lines 452-453: revision in response to Reviewer 1's comment: *[formerly] Line 34 [of SI] DIC in oceans surface is a mixture of air-sea gas exchange between surface and atmosphere as well as carbon exchange between surface and deep ocean. Look up references for the Marine calibration. See e.g. Stuiver, M. and T. F. Braziunas (1993). "Modeling atmospheric 14C influences and 14C Ages of Marine Samples to 10,000 BC." Radiocarbon 35: 137-189.*

Line 454: wording changes in response to Reviewer 1's comment: *[formerly] Line 35 [in SI] surely precipitation plays a role, however, what about land-use, catchment geochemistry etc These are important in delivering carbon to freshwater systems. On top of this carbon cycling and recycling within lakes may further be significant for some lake types.*

Line 457: minor wording change to reflect that we no longer have SI (advised by editors in AIP document)

Lines 460-463: minor wording change to reflect that we no longer have SI (advised by editors in AIP document); updated references (please see comment section of 'track changes' Word document for more details).

Line 476: response to Reviewer 1's comment: *[formerly] Line 56 [in SI] yes – I like that term 'regional' and please add the data from 47 and 48 to figure 3 and 4.* Please see comment section of 'track changes' Word document for details of our response

Line 483: acronym edit (advised by editors in AIP document)

Line 490: Heading change (advised by editors in AIP document)

Line 494-495: minor wording change to reflect that we no longer have SI (advised by editors in AIP document); typo fixed

Line 532: format change for consistency

Lines 541-542: minor wording change to reflect that we no longer have SI (advised by editors in AIP document)

Lines 548-550: update figure symbols and text associated with it in the Figure caption; reference error fixed

Line 552: format change for consistency

Line 558: acronym edit (advised by editors in AIP document)

Lines 562-565: Revision in response to Reviewer 1's comment: *[formerly] Fig 4 Out of curiosity: Do you have any ideas on why the data provides a negative t_0 ? Is it due to some unknown systematic error or is just noise?* Please see 'track changes' Word document for detailed response in the comment section; acronym edit (advised by editors in AIP document)

Line 576: Revision in response to Reviewer 1's comment: *[formerly] Fig 2ext please add a legend.* Legend added to Figure 8.

Lines 572, 579: acronym edits (advised by editors in AIP document)

Line 581: updated reference number

Lines 583-595: Revision in response to Reviewer 1's comment: *[formerly] [Extended] Fig 4 what does Hem (mean) denote – and how is it calculated? Perhaps add curve for north Atlantic too.* Figure caption revised and NW Atlantic curve added to Figure 9.

Lines 525-595 (general note): Figure captions consolidated (after including what was formerly SI and Extended Data, all into the same manuscript); a comment for each figure in the 'track changes' Word document denotes what figure numbers used to be (formerly)

Lines 619-622: minor wording change to reflect that we no longer have SI; footnote removed and added to Table caption (advised by editors in AIP document)

Line 639: wording change (as Reviewer 2's editorial change advised)

Line 641: acronym edit (advised by editors in AIP document)

Lines 665-815: formatting changes to references (as Reviewer 2's editorial changes advised)

Line 817: deleted because we do not have SI (advised by editors in AIP document)

Lines 821, 830-832: minor changes to Acknowledgements

Line 836: Specification added to Author contributions (advised by editors in AIP document and other publications in *Communications Biology*)